# Estimating the Long-Term Effects of Novel Treatments

**Keith Battocchi**[*]
Microsoft Research

**Eleanor W. Dillon**
Microsoft Research

**Maggie Hei**
Microsoft Research

**Greg Lewis**
Microsoft Research

**Miruna Oprescu**
Microsoft Research

**Vasilis Syrgkanis**
Microsoft Research

## Abstract

Policy makers often need to estimate the long-term effects of newly-developed treatments, while only having historical data of older treatment options. We propose a surrogate-based approach using a long-term dataset where only past treatments were administered and a short-term dataset where novel treatments have been administered. Our approach generalizes previous surrogate-style methods, allowing for continuous treatments and serially-correlated treatment policies while maintaining consistency and root-n asymptotically normal estimates under a Markovian assumption on the data and the observational policy. Using a semi-synthetic dataset on customer incentives from a major corporation, we evaluate the performance of our method and discuss solutions to practical challenges when deploying our methodology.

## 1 Introduction

Businesses invent new ways of interacting with their customers. Marketing departments devise new campaigns. Pharmaceutical companies roll out trials of new drugs. In many cases, decision makers expect these novel treatments to affect outcomes like sales or health over several years, but they are eager to begin evaluating these innovations within months of deployment.

We propose an estimation methodology that leverages historical data, collected under earlier treatment regimes, to derive early estimates of the long-term effects of novel, newly-deployed treatments. Athey et al. (2020) develop a surrogate index as one solution to this problem. This method relies on an assumption that causal effects on long-term outcomes are channeled through a set of observable short-term proxies. This assumption allows one to use the historical long-term data set to learn a mapping from short-term signals to a projected long-term reward and subsequently estimate the causal effect of novel treatments on this index of surrogates.

In practice, treatment policies are often *dynamic*: treatments are assigned repeatedly, and their assignments depend on past treatments and short-term outcomes. Unfortunately, this pattern can easily break the assumptions needed for the surrogate index to work. For example, suppose that a firm offers multiple investments to a particular customer in the historical data and these investments are auto-correlated, i.e. if a customer receives an investment this month, then they will receive an investment with high probability in one of the subsequent months. These future investments can substantially increase the long-term outcome of interest, and this increase will be attributed to the short-term proxies. The surrogate index thus formed will tend to over-predict long-run outcomes, and so when it is used to measure the treatment effect of some new treatment in the short-run data, the estimated treatment effect will be bigger (in absolute magnitude) than the truth.

---

[*]Correspondence to: Vasilis Syrgkanis <vasy@microsoft.com>.

35th Conference on Neural Information Processing Systems (NeurIPS 2021).

The main methodological innovation of this paper is to use the dynamic treatment effect analysis of Lewis & Syrgkanis (2020) on the historical data in order to create an unbiased, dynamically adjusted surrogate index. Our new index takes the interpretation of the projected long-term reward in the absence of any future treatments. This model produces unbiased causal effect estimates of the long-term effects of the novel treatments, even with auto-correlated treatments.

A second contribution of the paper is to generalize the final causal analysis step to allow for multiple continuous treatments rather than a single binary treatment, as in Athey et al. (2020). Our new estimator for this expanded surrogate approach allows for the construction of valid confidence intervals, even when using flexible machine learning models at both stages of the estimation to deal with high-dimensional data. In short, we show how one can combine three recently developed techniques, i) the surrogate index approach of Athey et al. (2020), ii) the double machine learning approach of Chernozhukov et al. (2018a) and iii) the dynamic treatment effect estimation approach of Lewis & Syrgkanis (2020), in a single data analysis pipeline to estimate treatment effects in the presence of dynamic treatment policies.

Our work lies in the broader field of estimating causal effects with machine learning and Neyman orthogonality (Neyman, 1979; Robinson, 1988; Ai & Chen, 2003; Chernozhukov et al., 2016; Chernozhukov et al., 2018b). Moreover, it relates to the work on machine learning estimation of treatment effects in the dynamic treatment regime (Nie et al., 2019; Thomas & Brunskill, 2016; Petersen et al., 2014; Kallus & Uehara, 2019b,a; Lewis & Syrgkanis, 2020; Bodory et al.; Singh et al., 2020) and on structural nested models in biostatistics (Robins, 1986; Robins et al., 1992; Robins, 1994; Robins & Ritov, 1997; Robins et al., 2000; Lok & DeGruttola, 2012; Vansteelandt et al., 2014; Vansteelandt & Sjolander, 2016). Finally, it relates to the surrogacy literature in causal inference (Prentice, 1989; Begg & Leung, 2000; Frangakis & Rubin, 2002; Freedman et al., 1992). In the following sections we build on insights in these works to design a new, streamlined method that resolves several difficult aspects of a common applied problem.

## 2 Problem and Methodology

We illustrate the difficulties of our problem and how we our method resolves them with a running example of a firm making investments in its customers in order to increase subsequent purchases. In this section we describe the problem set up and data requirements and use a simple two-period model to build intuition for how our approach estimates long-term causal effects in this environment and why earlier approaches fail. In the following section we provide formal proofs of our methodology in the general, multi-period case.

### 2.1 Problem statement

A firm has a number of distinct treatments $T_1, T_2 \ldots T_k$ it can offer to customers, such as discounts or access to special services. At each period $t$ (say, a month), a vector of treatments $T_{i,t} = (T_{i,t,1}, \ldots, T_{i,t,k}) \in \mathbb{R}^k$, is applied to each customer $i$. We also observe a vector of $p$ characteristics $X_{i,t} = (X_{i,t,1}, \ldots, X_{i,t,p}) \in \mathbb{R}^p$, some of which are constant within customer (e.g. industry) and some of which vary over time and customer (e.g. last month's revenue). We observe the outcome of interest $Y_{i,t} \in \mathbb{R}$ (e.g. monthly revenue).

Treatments may affect outcomes over many future periods, and this horizon could vary by treatment. We are therefore interested in identifying the average effect of each treatment at some period $t$, on the cumulative outcome in the subsequent $M$ periods, i.e. $\bar{Y}_{i,t} = \sum_{\kappa=1}^{M} Y_{i,t+\kappa}$. Suppose direct inference is impossible because we have not yet observed the subsequent $M$ periods following the introduction of a new treatment. Instead, we have access to a vector of $d$ short-term proxies/surrogates, $S_{i,t} = (S_{i,t,1}, \ldots, S_{i,t,d})$ that act as leading indicators for our target outcome. In this example, we would like surrogates that capture a customer's trajectory such as the next few months of revenue, intensity of product usage, new contract commitments, and participation in company events.

### 2.2 Dynamic Adjusted Surrogate Index

The key innovation of all surrogate-based solutions is to use the proxy variables to bridge two incomplete data sets to estimate $\bar{Y}_{i,t}$. A long-term *observational data set* need only include customer characteristics $X_{i,t}$, surrogates $S_{i,t+1}$, realized $M$-period outcomes $\bar{Y}_{i,t}$. A second short-term,

*experimental data set*, includes customer features, $X_{i,t}$, surrogates, $S_{i,t}$, and treatments $T_{i,t}$. This experimental data set is restricted to the period where all treatments of interest have been introduced, but only requires a few months of leading surrogates rather than $M$ periods of leading outcomes. We propose the following estimation strategy:

1. Use the observational data set to estimate the expected long-term outcome net of any post-period $t$ treatments using the recursive methodology of Lewis & Syrgkanis (2020). This adjusted outcome, $\bar{Y}_{it}^{adj}$, is the expected cumulative outcome starting in period $t$ under the assumption that customer $i$ receives no further treatments over the next $M$ periods.

2. Still using the observational data set, construct a model that predicts this adjusted outcome from $S_{i,t}$ and $X_{i,t}$:

$$g_0^{adj}(S_{i,t}, X_{i,t}) := \mathbb{E}[\bar{Y}_{i,t}^{adj} \mid S_{i,t}, X_{i,t}]. \qquad \text{(surr. model)}$$

Any machine learning estimation algorithm can be used to construct $\hat{g}$.

3. Finally, calculate the adjusted surrogate index using the surrogates in the experimental data set:

$$I_{i,t} := \hat{g}(S_{i,t}, X_{i,t}). \qquad \text{(predicted surrogate index)}$$

This predicted index becomes the outcome in the final causal model.

This approach will be consistent for the treatment effects under three assumptions, which we illustrate as a causal graph in Figure 1.

1. **Conditional Independence**: There are no paths from current treatments to future outcomes except through current or future surrogates. Formally, $T_{it} \perp Y_{is} | X_{it}, S_{it}..S_{is}$ for all $s \geq t$.

2. **Stability**: Relationships between current and future surrogates and between surrogates and outcomes, marked in red in the figure, are constant over time and across samples.

3. **Unconfoundedness**: There are no variables beyond our observed set that directly affect both the treatment assignment and outcome.

Note that the first assumption allows relationships between current and future treatments and between current treatments and future surrogates, in contrast to the canonical surrogate model. The stability assumption does not extend to the relationships between treatments and surrogates, which allows for the introduction of novel treatments in the experimental data not found in the observational data. The model is able to account for steady growth in a natural way: as the surrogates grow, the predicted outcomes grow too.

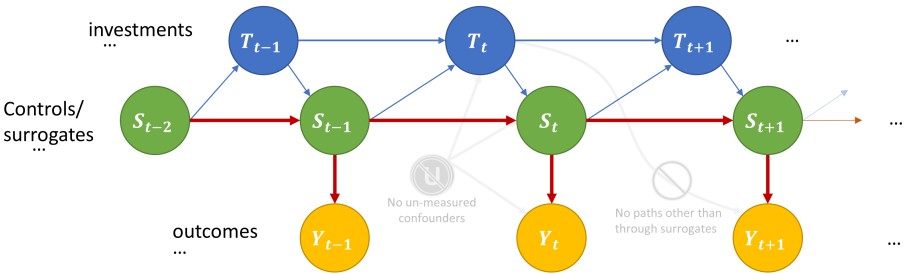

Figure 1: Causal graph representation of the main assumptions of our causal analysis.

## 2.3 A two-period example

For conciseness, we now collapse $X_{i,t}$ and $S_{i,t-1}$ so that the surrogates at each period and the controls in the next period are all denoted by $S_{i,t-1}$. In other words, all customer observable characteristics

in the next period are candidate surrogates and become controls for the period after next. This is without loss of generality. For the remainder of this section only, we further simplify by assuming a linear Markovian model of how the random variables evolve over time and focusing on a single scalar investment ($k = 1$). With these simplifications, the evolution of the random variables can be described in three equations:

$$S_{i,t} = AT_{i,t} + BS_{i,t-1} + \epsilon_{i,t} \qquad \text{(evolution of controls)}$$
$$Y_{i,t} = \gamma' S_{i,t} + \zeta_{i,t} \qquad \text{(outcome equation)}$$
$$T_{i,t+1} = \kappa T_{i,t} + \lambda' S_{i,t} + \eta_{i,t} \qquad \text{(treatment policy)}$$

where $A$ is an $p \times 1$ matrix, $B$ is a $p \times p$ matrix, $\gamma, \lambda$ are $p$-dimensional vectors and $\kappa$ is a scalar. The terms $\epsilon_{i,t}, \zeta_{i,t}, \eta_{i,t}$ are exogenous mean-zero independent noise terms. For conciseness we drop the customer index $i$ in the remaining equations.

**Target effect estimand**    With two periods, the long term outcome $\bar{Y}_{i,t} = Y_{i,t} + Y_{i,t+1}$. The effect of treatment $T_t$ on $\bar{Y}_t$ can be derived from the structural model equations:

$$\bar{Y}_t = \gamma' S_t + \gamma' S_{t+1} + \text{noise} = \gamma' AT_{t+1} + \gamma'(B+I)AT_t + \gamma'(B+I)BS_{t-1} + \text{noise}^2$$

Thus we see that the effect of $T_t$ on $Y_{t+1} + Y_t$, assuming future treatments are held constant, is:

$$\theta_0 := \gamma'(B+I)A. \tag{1}$$

**Bias from a standard surrogate index**    Athey et al. (2020) propose training a surrogate index using the realized cumulative outcome $\bar{Y}_t$ on $S_t$, essentially our proposed step 2 using raw instead of adjusted outcomes. Following that approach, we would estimate:

$$g_0(S_t) = \mathbb{E}[\bar{Y}_t \mid S_t] = \gamma'(I+B)S_t + \gamma' A\mathbb{E}[T_{t+1} \mid S_t] \tag{2}$$

Subsequently, if we estimate the causal effect of $T_t$ on $g_0(S_t)$, then this effect would be:

$$\theta_* = \theta_0 + \gamma' A\frac{\mathbb{E}[\mathbb{E}[T_{t+1} \mid S_t] T_t]}{\mathbb{E}[T_t^2]} \tag{3}$$

The standard estimate contains a bias stemming from the fact that investment today can lead to higher investment tomorrow. As illustrated in equation above, this bias can stem from two channels, both of which are plausible in many use cases. In our example, a customer that is treated once may build a stronger relationship with the firm and receive further treatments (i.e. $\kappa \geq 0$ in the treatment policy). Alternatively, if the firm targets treatments to fast-growing customers, any treatment that successfully drives surrogate metrics higher will make the customer eligible for further treatment ($\lambda' A \geq 0$).

**Dynamic adjustment**    With only two periods, our first step simply requires estimating the single-period causal effect of $T_{t+1}$ on $Y_{t+1}$, controlling for $S_t$. This conditional expectation is equal to:

$$\mathbb{E}[Y_{t+1} \mid T_{t+1}, S_t] = \gamma' AT_{t+1} + \gamma' BS_t. \tag{4}$$

We can now create an *adjusted long-term outcome*, $\bar{Y}_t^{\text{adj}} := Y_t + Y_{t+1} - \alpha'_{t+1}T_{t+1}$, where $\alpha_{t+1}$ is our estimate of $\gamma' A$ from this first step.

The *dynamically adjusted surrogate index* builds this adjustment into equation (2) above:

$$\begin{aligned} g_0^{\text{adj}}(S_t) &= \mathbb{E}[Y_t \mid S_t] + \mathbb{E}[Y_{t+1} - \gamma' AT_{t+1} \mid S_t] \\ &= \gamma' S_t + \mathbb{E}[\gamma' BS_t \mid S_{t+1}] \\ &= \gamma'(I+B)S_t \end{aligned} \tag{5}$$

This new index captures the projected $M$-period outcomes as if the customer was offered no future treatments.

When we estimate the effect of the treatment based on this adjusted surrogate index we recover:

$$\begin{aligned} \mathbb{E}[g_0^{\text{adj}}(S_t) \mid T_t, S_{t-1}] &= \gamma'(I+B)\mathbb{E}[S_t \mid T_t, S_{t-1}] \\ &= \gamma'(I+B)AT_t + \gamma'(I+B)BS_{t-1} \end{aligned} \tag{6}$$

With the dynamic adjustment, the coefficient in front of $T_t$ that we recover is the true causal effect $\theta_0 = \gamma'(I+B)A$. Lewis & Syrgkanis (2020) show how this adjustment approach can be extended to many periods, via a recursive peeling process, and also to high-dimensional surrogates and controls via a dynamic double machine learning approach.

**Estimating causal effects of new treatments**   With this adjusted surrogate index in hand, we can also estimate the long-term effect of any other treatment that was introduced more recently. In this example, the stationarity assumption of the proposed surrogate approaches requires that $B$, which governs how the surrogate evolves, and $\gamma$, which governs how surrogates translate to per-period outcomes, do not change between the observational and experimental data-sets. These two parameters govern how surrogates today relate to future outcomes in the absence of any treatment.

Under such a condition, if we introduce a new treatment $T_t^{\text{new}}$, which has a different effect $A^{\text{new}}$ on the surrogates (and hence on the long-term outcome), then the effect of this treatment on the long-term outcome is $\theta_0^{\text{new}} = \gamma'(I + B)A^{\text{new}}$. This $\theta_0^{\text{new}}$ is exactly the outcome of estimating the causal effect of $T_t^{\text{new}}$ on $g_0^{\text{adj}}(S_{t+1})$ controlling for $S_t$.

# 3   Formal Guarantees

In this section we present our main results. In addition to expanding to $M$ periods, we introduce two substantive innovations over the basic strategy presented in the two period example above. First, we develop a generalization of the doubly robust estimation method in Athey et al. (2020) to the case of multiple continuous treatments. Second, we relax the linearity assumptions in the structural model and make use of orthogonal machine learning techniques Chernozhukov et al. (2018a) to allow for a rich set of potential confounders and valid analytic confidence intervals.

We now assume that we observe the full $M$-period time series, $(S_0, T_1, S_1, Y_1, T_2, S_2, Y_2, \ldots, T_M, S_M, Y_M)$, for each sample in the observational popula-tion, denoted as $o$. In contrast, we only observe $(S_0, T_1, S_1)$ for each sample from the experimental population, denoted as $e$.

**Notation**   Throughout, we will denote with $\mathbb{E}_e[\cdot]$ the expectation conditional on the experimental population and $\mathbb{E}_o[\cdot]$ the expectation conditional on the observational population. Moreover, for any vector-valued function $f$ that takes as input a random variable $Z$, we denote with:

$$\|f\|_{2,o} = \sqrt{\mathbb{E}_o\left[\|f(Z)\|_2^2\right]} \tag{7}$$

and analogously $\|f\|_{2,e}$. We denote with $\mathbb{E}_n[\cdot]$, the empirical expectation over all the samples, i.e. for any random variable $Z$, $\mathbb{E}_n[Z] := \frac{1}{n}\sum_i Z_i$, and with $\mathbb{E}_{e,n}$ and $\mathbb{E}_{o,n}$ the empirical expectation over the experimental and observational samples correspondingly, i.e. $\mathbb{E}_{e,n}[Z] = \frac{1}{n_e}\sum_{i \in e} Z_i$ and $\mathbb{E}_{o,n}[Z] = \frac{1}{n_o}\sum_{i \in o} Z_i$.

Our goal is to isolate the causal effect of treatments in period 1, $T_1$, on the long term outcome $\bar{Y}_1 := \sum_{t=1}^M Y_t$. Formally, if we fix the future treatments that each sample receives and we change the treatment $T_1$ from some value $t_0$ to some other value $t_1$, the treatment effect is:

$$\tau(t_1, t_0) := \mathbb{E}_e[\mathbb{E}[\bar{Y}_1 \mid do(T_1 = t_1), T_2, \ldots, T_M] - \mathbb{E}[\bar{Y}_1 \mid do(T_1 = t_0), T_2, \ldots, T_M]]. \tag{8}$$

We present our theoretical results in two steps. For exposition, we begin by assuming treatments happen only at period 1. This is the setting analyzed in Athey et al. (2020), albeit only for the case of a single binary treatment $T_1$.[3] We then show how this approach can be modified to incorporate a dynamic treatment policy in the observational and experimental sample.

## 3.1   Double/Debiased Correction without Dynamic Adjustment

To begin, we assume that $T_2, \ldots, T_M = 0$. Throughout, we assume a *partially linear relationship* between the treatment and the long-term outcome:

$$\mathbb{E}_e[\bar{Y}_1 \mid T_1, S_0] = \theta_0^\top \phi(T_1, S_0) + b_0(S_0) \tag{PLR}$$

---

[3]Athey et al. (2020) also allowed for estimation of average treatment effects, even in the case when there is arbitrary treatment effect heterogeneity. In this work, we assume that treatment effects are constant. A generalization to the case of arbitrary treatment effect heterogeneity is feasible, but would require the estimation of conditional covariance matrices, which would make the estimation algorithm more brittle and the exposition much more complex.

for some known feature map $\phi(\cdot, \cdot)$, but arbitrary function $b_0(\cdot)$.

Formally, the *invariance of the surrogate-outcome* relationship requires that that the mean-relationship between the surrogates $S_1$ and the long-term outcome does not change between the observational and the experimental sample:

$$g_0(S_1) := \mathbb{E}_o[\bar{Y}_1 \mid S_1] = \mathbb{E}_e[\bar{Y}_1 \mid S_1]. \tag{IR}$$

Under the PLR assumption and the causal graph governing our data, we have that:

$$\tau(t_1, t_0) = \theta_0^\top \mathbb{E}[\phi(t_1, S_0) - \phi(t_0, S_0)]. \tag{9}$$

We establish three consistent estimators for $\theta_0$ that follow the same intuition as Athey et al. (2020), adapted to consider linear effects of continuous treatments rather than a single binary treatment.

**Theorem 3.1** (Identification). *Denote the residual surrogate index and the residual treatment with:*

$$\tilde{g}_0(S_1) := g_0(S_1) - \mathbb{E}_e[g_0(S_1) \mid S_0] \tag{10}$$

$$\tilde{T}_1 := \phi(T_1, S_0) - \mathbb{E}_e[\phi(T_1, S_0) \mid S_0] \tag{11}$$

*Then under assumptions (PLR) and (IR):*

$$\theta_0 = \mathbb{E}_e[\tilde{T}_1 \tilde{T}_1^\top]^{-1} \mathbb{E}_e[\tilde{T}_1 \tilde{g}_0(S_1)] \qquad\qquad \text{(surrogate index rep.)}$$

$$\theta_0 = \mathbb{E}_e[\tilde{T}_1 \tilde{T}_1^\top]^{-1} \mathbb{E}_o\left[\frac{\Pr(e \mid S_1, S_0)}{\Pr(o \mid S_1, S_0)} \frac{\Pr(o)}{\Pr(e)} \mathbb{E}_e[\tilde{T}_1 \mid S_1, S_0] \bar{Y}_1\right] \qquad \text{(surrogate score rep.)}$$

$$\theta_0 = \mathbb{E}_e[\tilde{T}_1 \tilde{T}_1^\top]^{-1} \left(\mathbb{E}_e[\tilde{T}_1 \tilde{g}_0(S_1)] + \mathbb{E}_o\left[\frac{\Pr(e \mid S_1, S_0)}{\Pr(o \mid S_1, S_0)} \frac{\Pr(o)}{\Pr(e)} \mathbb{E}_e[\tilde{T}_1 \mid S_1, S_0](\bar{Y}_1 - g_0(S_1))\right]\right)$$

$$\text{(orthogonal rep.)}$$

The core estimation challenge that the surrogate approach resolves is that the treatments and outcome of interest are not observed in a single dataset. Intuitively, the first **surrogate index representation** approaches this challenge by using realized treatments from the experimental sample and, in place of realized outcomes, substitutes the expected outcome conditional on the surrogates, $g_0(S)$, which can be identified from the observational sample and then constructed in the experimental sample.

The second **surrogate score representation** reverses this substitution. The second term pairs an expectation of the treatment conditional on surrogates, $\mathbb{E}_e[\tilde{T}_1 \mid S_1, S_0]$, with the realized outcomes from the observational sample. This representation requires an added ratio of probabilities of appearing in each sample, $\Pr(e)$ and $\Pr(o)$, to adjust for sampling variation across the two datasets.

The third **orthogonal representation** blends the first two representations and satisfies *Neyman orthogonality*, which allows the construction of confidence intervals and double robustness.

The parameter identified in the equations in Theorem 3.1 is interpretable even if the partially linear assumption is violated. In this case, the equations are identifying the best linear projection of the variation in the long-term outcome that is not explained by the initial state. We formulate the estimation of $\theta_0$ based on the third orthogonal representation as a $Z$-estimator based on a vector of moment equations that depends on a vector of nuisance functions $f_0$, i.e.:

$$m(\theta_0; f_0) := \mathbb{E}[\psi(Z; \theta_0, f_0)] = 0 \tag{12}$$

and such that it satisfies the Neyman orthogonality condition:

$$D[m(\theta_0; f_0), f - f_0] := \left.\frac{\partial}{\partial t} m(\theta_0; f_0 + t(f - f_0))\right|_{t=0} = 0$$

Subsequently, this will allow us to invoke the results in Chernozhukov et al. (2018b), to derive an asymptotic normal estimator, even when high-dimensional, regularized approaches are used to estimate the nuisance functions $f_0$.

**Theorem 3.2** (Orthogonal Moment Formulation). *Let $f = (g, q, p, h)$, denote a set of nuisance functions and define:*

$$q_0(S_1, S_0) := \frac{\Pr(e \mid S_1, S_0)}{\Pr(o \mid S_1, S_0)} \mathbb{E}_e[\tilde{T}_1 \mid S_1, S_0]$$

$$p_0(S_0) := \mathbb{E}_e[\phi(T_1, S_0) \mid S_0]$$

$$h_0(S_0) := \mathbb{E}_e[g(S_1) \mid S_0]$$

*Then $\theta_0$ is the solution to the moment equation:*

$$m(\theta_0; f_0) := m_e(\theta_0; f_0) + m_o(\theta_0; f_0) = 0$$

$$m_e(\theta; f) := \mathbb{E}\left[1\{e\}\left(g(S_1) - h(S_0) - \theta^\top(\phi(T_1, S_0) - p(S_0))\right) \cdot (\phi(T_1, S_0) - p(S_0))\right]$$

$$m_o(\theta_0; f) := \mathbb{E}\left[1\{o\}\, q(S_1, S_0)(\bar{Y}_1 - g(S_1))\right]$$

*Moreover, the moment $m$ satisfies the Neyman orthogonality property with respect to $f$. Furthermore, it satisfies a stronger double robustness property with respect to $g$ and $q$, i.e. if we denote with $\hat{\theta}$ the solution to $m(\theta; \hat{g}, \hat{q}, p_0, \hat{h}) = 0$, then:*

$$\left\| \mathbb{E}_e[\tilde{T}_1 \tilde{T}_1^\top]\left(\hat{\theta} - \theta_0\right) \right\|_2 \leq \frac{\Pr(o)}{\Pr(e)} \|g_0 - \hat{g}\|_{2,o} \|q_0 - \hat{q}\|_{2,o}$$

We estimate $\theta_0$ by method of moments, solving the empirical analogues of these moment conditions, after estimating the nuisance functions $f$. We show that this estimator is asymptotically normal under relatively mild rate conditions on the estimation of the nuisances in Appendix Theorem A.2 (Theorem 3.3, below, establishes the same result for a more general case).

### 3.2 Double/Debiased Correction with Dynamic Adjustment

We now consider the case of potentially serially correlated treatments in all periods. To achieve consistent estimates of the effects of period 1 treatments in this setting we need to again assume that all the relationships are linear, albeit potentially high-dimensional:

$$\begin{aligned}
S_t &= AT_t + BS_{t-1} + \epsilon_t \\
Y_t &= CS_t + \eta_t \\
T_{t+1} &= DT_{t-1} + GS_{t-1} + \zeta_t
\end{aligned} \tag{13}$$

where $\epsilon_t, \eta_t, \zeta_t$ are i.i.d. random shocks.

In this case, the main identification result in Lewis & Syrgkanis (2020), shows that:

$$\tau(t_1, t_0) = \sum_{t=1}^{M} \theta_{1,t}^\top (t_1 - t_0) = \theta_0^\top (t_1 - t_0) \tag{14}$$

where $\theta_0 := \sum_{t=1}^{M} \theta_{1,t}$ and $\theta_{1,t}$ are interpreted as the dynamic effect of treatment at period 1 on the outcome at period $t$, controlling for all future treatments. This quantity is equivalent to the effect assuming that all future treatments are zero, since the effects of each period treatment under this linear model are linearly separable. This property simplifies both the estimation and the interpretation of the effect quantity as many causal quantities of potential interest collapse in this setting to the same object.

The identification results in Lewis & Syrgkanis (2020) also imply that we can identify the quantity $\theta_{1,t}$ by estimating the effect of $T_1$ on the multi-period dynamically adjusted target outcome:

$$\bar{Y}_1^{\text{adj}} = \sum_{t=1}^{M}\left[Y_t - \sum_{\tau=2}^{t} \theta_{\tau,t}^\top T_t\right] = \theta_0^\top T_1 + \beta_0 S_0 + \mu_1 \tag{15}$$

for some exogenous mean-zero random shock variable $\mu_1$. This is the setup that we analyzed in the previous section, albeit with $\bar{Y}_1$ replaced by $\bar{Y}_1^{\text{adj}}$ and with $\phi(T_1, S_0) = T_1$. The PLR assumption with respect to this dynamically adjusted target immediately holds by the equation above.

Furthermore, we need to assume that the invariance of the surrogate-outcome assumption holds, albeit now for the dynamically adjusted long-term outcome:

$$g_0^{\text{adj}}(S_1) := \mathbb{E}_o[\bar{Y}_1^{\text{adj}} \mid S_1] = \mathbb{E}_e[\bar{Y}_1^{\text{adj}} \mid S_1] \tag{dynIR}$$

This assumption is more permissive in practice than the standard invariance assumption as we no longer require that the dynamic treatment policy in the observational data be the same as in the

experimental data, but only that the adjusted outcomes retain the same relationship with the surrogates. For estimation, we can apply the same analysis as in the previous section. The differences are that 1) we now have this new target long-term outcome and 2) we must additionally account for the variance of the $\hat{\theta}_t$ terms that are being estimated by the dynamic adjustment algorithm. We articulate the full algorithm including these additions in Appendix B. We show below that the method of moments estimator with plug in nuisance estimates is asymptotically normal:

**Theorem 3.3** (Surrogates with Dynamic Adjustment: Estimation and Asymptotic Normality). *Let:*

$$\psi_0(Z;\theta,f_0) := 1\{e\} \underbrace{\left(\tilde{g}_0^{adj}(S_1) - \theta_0^\top \tilde{T}_1\right)\tilde{T}_1}_{\psi_{0,e}(Z;\theta,f_0)} + 1\{o\} \underbrace{\left(\bar{Y}_1^{adj} - g_0^{adj}(S_1)\right)q_0(S_1,S_0)}_{\psi_{0,o}(Z;\theta,f_0)}$$

$$\forall\, 2 \leq \tau \leq t \leq M : \psi_{\tau,t}(Z;\theta,f_0) := 1\{o\} \left(\tilde{Y}_{t,\tau} - \sum_{\kappa=\tau+1}^{t}\theta_{\kappa,t}^\top \tilde{T}_{\kappa,\tau} - \theta_{\tau,t}^\top \tilde{T}_{\tau,\tau}\right)\tilde{T}_{\tau,\tau}$$

$$\forall\, 2 \leq t \leq M : \psi_t(Z;\theta_0,f_0) = [\psi_{t,t}(Z;\theta,f_0) \quad \ldots \quad \psi_{t,2}(Z;\theta,f_0)]^\top$$

*where we use the short-hand notation $\tilde{Y}_{t,\tau} = Y_t - \mathbb{E}[Y_t \mid S_{\tau-1}]$ and $\bar{Y}_1^{adj} = \sum_{t=1}^{M}\left(Y_t - \sum_{\tau=2}^{t}\theta_{\tau,t}^\top T_\tau\right)$ and $\tilde{T}_{t,\tau} = T_t - \mathbb{E}[T_t \mid S_{\tau-1}]$. Let $h_{t,\tau}(S_{\tau-1}) = \mathbb{E}[Y_t \mid S_{\tau-1}]$ and $p_{t,\tau}(S_{\tau-1}) = \mathbb{E}[T_t \mid S_{\tau-1}]$. Let $f = \{g,q,p,h,\{h_{\tau,t},p_{\tau,t}\}_{2\leq\tau\leq t\leq M}\}$, denote the set of all nuisance functions and let $f_0$ denote their true value. Consider the estimator based on the empirical version of the orthogonal moment, with plug-in nuisance estimates $\hat{f}$ trained on a separate sample, i.e. $\hat{\theta}$ solves the empirical moment equation:*

$$m_n(\hat{\theta};\hat{f}) = 0 \qquad m_n(\theta;f) := \mathbb{E}_n[\psi(Z;\theta,f)] := \begin{bmatrix} \mathbb{E}_n\left[\psi_0(Z;\theta,f)\right] \\ \mathbb{E}_n\left[\psi_2(Z;\theta,f)\right] \\ \ldots \\ \mathbb{E}_n\left[\psi_M(Z;\theta,f)\right] \end{bmatrix} \qquad (16)$$

*If $\|\hat{r} - r_0\|_{2,o} = o_p(n^{-1/4})$, for $r \in \{\{h_{\tau,t},p_{\tau,t}\}_{2\leq\tau\leq t\leq M}\}$ and if $\|\hat{h} - h_0\|_e = o_p(n^{-1/4})$, $\|\hat{p} - p_0\|_e = o_p(n^{-1/4})$ and $\|g_0 - \hat{g}\|_{2,o} \cdot \|q_0 - \hat{q}\|_{2,o} = o_p(n^{-1/2})$, then:*

$$\sqrt{n}\left(\hat{\theta}_0 - \theta_0\right) \to_d N(0, J_e^{-1}\left(\Sigma_e + \Sigma_{o,1} + \Sigma_{o,2}\right)J_e^{-1})$$

*where $J_e, \Sigma_e, \Sigma_{o,1}, \Sigma_{o,2}$ are defined in the appendix.*

## 4 Semi-Synthetic Experimental Evaluation

To evaluate the performance of our proposed estimation strategy we construct a semi-synthetic dataset that retains qualitative characteristics of data on real-world incentive investments in customers at a major corporation, while preserving confidentiality. The semi-synthetic data set preserves several common patterns that require thoughtful attention. The treatments, in this case incentive investments, are lumpy: in most periods most customers get no investments. Proxies, which include single period values of the outcome of interest, are highly auto-correlated over time. Treatments are also auto-correlated, and correlated with past values of proxies. The data also include time-invariant controls that affect proxies and treatments in all periods.

To build the data we estimate a series of moments from a real-world dataset: a full covariance matrix of all proxies, treatments, and controls in one period and a series of linear prediction models (lassoCV) of each proxy and treatment on a set of 6 lags of each treatment, 6 lags of each proxy, and time-invariant controls. Using these values, we draw new parameters from distributions matching the key characteristics of each family of parameters. Finally, we use these new parameters to simulate proxies, treatments, and controls by drawing a set of initial values from the covariance matrix and forward simulating to match intertemporal relationships from the transformed prediction models. Finally, we arbitrarily select one proxy to be the outcome of interest and construct the cumulative sum of this selected outcome over four or eight periods. For further details on the data generation process, see the appendix.

The true treatment effects in the synthetic data are known functions of parameters from the linear prediction models. We then compare these true causal effects to estimated effects using a variety of

approaches, gradually incorporating our proposed innovations. Figure 2 shows the distribution of the $\ell_2$ estimation error ($\|\hat{\theta} - \theta_0\|_2$) for each approach over 100 simulated data sets. The top row plots the estimation error when estimating the effect on four periods of outcomes, increasing the sample size of each simulation from left to right, while the bottom row shows the same for the effect on eight periods of outcome.

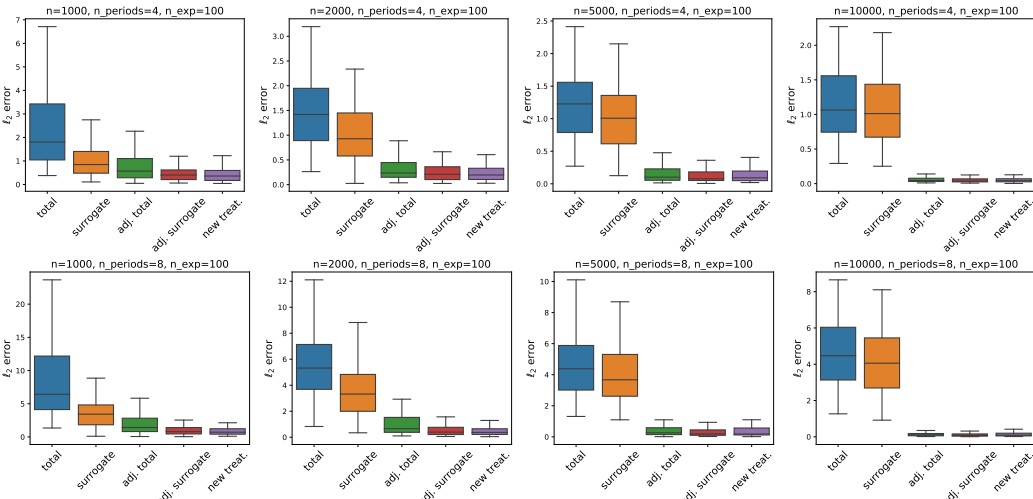

Figure 2: Experimental performance for $M = 4$ periods and $M = 8$ periods.

Because we construct a single, long synthetic data set for this experiment it is possible to estimate the treatment effects on realized long-term outcomes directly, unlike the typical use case for a surrogate approach. We begin by estimating the effect of each treatment at time $t$ directly on the realized total outcomes, rather than the surrogate index, without dynamic adjustment. The blue, "total" bars in each panel of Figure 2 display substantial bias from this method, which does nothing to control for *future* treatments. Because current treatments are positively correlated with future treatments in the semi-synthetic data, this first approach consistently overestimates treatment effects.

We then estimate the same set of treatment effects using the surrogate approach described in Section 3.1, still without dynamic adjustment. The distribution of estimation errors from this approach is represented in the orange "surrogate" bars. The estimated treatment effects are still substantially larger than the true effects on average, but the surrogate model exhibits slightly less bias than the direct "total" approach. Intuitively, because the surrogate approach is only capturing the relationship between treatment and outcome that passes through the surrogates it picks up less of the bias resulting from future correlated treatments than the direct approach.

The third set of green "adj. total" bars plot errors when estimating treatment effects on adjusted realized outcomes using the method of Lewis & Syrgkanis (2020), the preferred approach when all treatments of interest and realized long-run outcomes are available in a single data set. This method removes the effects of future treatments from the long-run outcome in a first step and, as expected, exhibits significantly less bias than the first two methods, particularly for reasonably large samples.

The final two bars illustrate the success of the adjusted surrogate approach described in Section 3.2. As illustrated by the red "adj. surrogate" bars, our proposed approach is highly accurate in predicting long-term effects with a performance comparable to that of having access to the raw long-term outcome itself. The final purple "new treat." bars show that the approach works equally well when considering the effect of a novel treatment that appears only in the experimental sample and was not part of the dynamic adjustment. Overall, this methodology overcomes a common data limitation when considering long-term effects of novel treatments and expands the surrogate approach to consider a common, and previously problematic, pattern of serially correlated treatments.

**Acknowledgments and Disclosure of Funding**

The authors have no sources of additional funding to disclose.

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
