# OpenReview forum: "Estimating the Long-Term Effects of Novel Treatments"
_NeurIPS.cc/2021/Conference — NeurIPS 2021 Poster_

### Official Review · Reviewer_5T1d · 2021-07-08

**Rating:** 6
**Confidence:** 3

**Summary:**

This paper tackles the problem of estimating the long-term effect of some new treatment in the case where the treatment assignment is decided adaptively. Athey et al.(2020) proposed the long-term effect estimation for the stable treatment assignment case. This paper applied the method proposed in Lewis & Sygkanis(2020) to adjust the approach of Athey et al.(2020) for the case where the treatment assignment is decided adaptively.

**Limitations And Societal Impact:**

They adequately addressed the limitation in their paper.

**Main Review:**

Estimating the long-term effect is an important problem in the application of causal inference and econometrics. However, I am quite not sure that this research is well motivated. Do we really need to know the long-term effect of a particular arm while using some dynamic treatment regime? I appreciate it if you could explain and add some examples.

For the experimentation, you should report your result of the synthetic data experiment reported in the appendix with more detailed information about the data generating process.

Some minor issues

- line 121: what is "noise" in the equation?
- line 121: the definition of I is ambiguous here
- section 3.2: the actual estimation step should be explained here. At least, you should refer to the appendix here.
- line 280: "for this excercise," ?
- line 282: the definition of "total" is somewhat ambiguous. For me, the definition in the appendix helped.

**Time Spent Reviewing:**

8

---

> ### Author Response · Authors · 2021-08-10
> **Review response**
>
> We thank the reviewer for the elaborate review and comments and address each one in detail:
>
> 1) Our research was motivated by a real-world problem that we solved in a major corporation, where we truly wanted to know the long-term effect of an investment that we make today on the two year revenue from a customer. Long-term effects are really the quantity of interest from a business strategy perspective. But business decisions need to be made at much faster pace than the horizon that we care about (e.g. need to know which investment programs to de-prioritize or cease every 6 months). Moreover, even if you were to deploy more treatments in the future, knowing how much to attribute to the action you just gave is also very important from an attribution point of view, since you want to understand which arms to de-prioritize or cease to offer (e.g. cease to offer particular discount programs). In the particular setting that we deployed the method, you can view our target quantity we estimate, as exactly the long-term effect that should be attributed to a novel investment that we deployed in the last 6 months on 2 year revenue, so that we can make a decision whether to continue it or not. The surrogate approach allows us to do that, given that we have a lot of historical data from other customers. And the dynamic treatment problem in the historical data is a real-world issue that needs to be dealt with.
>
> 2) We will clarify on the use of “noise”. Essentially “noise” contains ugly combination of epsilon, zeta and eta noise terms and we wanted to avoid complicated expressions in this expository part. I is the identity matrix. We will add a footnote.
>
> 3) We will add a reference to the appendix and a short algorithmic description summarizing it.
>
> 4) “exercise” : we will replace with “experiment”
>
> 5) We will clarify the “total” method using content from the appendix description.

---

> > ### Comment · Reviewer_5T1d · 2021-08-16
> > **reply**
> >
> > Thank you for your clarification!
> >
> > I still wonder about the gap of the experimental result between fully-synthetic and semi-synthetic data. Do you have any potential reason for such a difference?

---

> > > ### Author Response · Authors · 2021-08-16
> > > **synthetic vs semi-synthetic**
> > >
> > > Are you referring to the difference in the magnitude of errors in the appendix figures, between the synthetic and semi-synthetic in the Appendix? If so then there are several reasons why the absolute magnitude is different:
> > > 1) the noise at each time step of the time series and the initial random variables in the synthetic data is generated via a simple gaussian distribution. On the other hand the noise in the semi-synthetic data is generated via a heavy tailed distribution which matches several of the moments and qualitative characteristics of the unobserved variation we found in the real data.
> > > 2) the autocorrelation is more simple in the synthetic example, while in the semi-synthetic it stems from predictive models that we fitted to the real data.
> > >
> > > Both of these main differences, lead to qualitatively different data degenerating processes. This can lead for instance to larger magnitude "noise" in the semi-synthetic data, which leads to large mean squared error. Moreover, the heavy tails in the semi-synthetic data is a genuinely harder setting.
> > >
> > > Despite this, we observe the same qualitative trends among the different methods in both settings, and we find that the error of our methods goes down to zero as sample size grows in both settings. Albeit in the semi-synthetic these errors will be slightly larger pointwise, as we are dealing with more "noisy" data.

---

> > > > ### Comment · Reviewer_5T1d · 2021-08-25
> > > > **reply**
> > > >
> > > > Thank you for your explanation.
> > > > I appreciate it if you could add this comparison into your appendix.

---

> > > > > ### Author Response · Authors · 2021-08-25
> > > > > **Updating appendix**
> > > > >
> > > > > We will definitely add a comparison explaining the differences between the semi-synthetic and synthetic in the appendix and the reason for the differences in performance. Thank you for the suggestion!

---

### Official Review · Reviewer_JAeM · 2021-07-16

**Rating:** 6
**Confidence:** 4

**Summary:**

The paper considers the problem of estimating a long-term treatment effect. It considers the setting where there are two datasets: an observational dataset, which contains the long-term outcome, the treatment and the surrogate variable, and an experimental dataset, which contains only the treatment and the surrogate variable. The authors generalize the surrogate index method proposed by Athey et al. (2020), by considering dynamic treatment policies. Their new proposed method is a combination of the surrogate index approach, the double machine learning approach and the dynamic treatment effect estimation approach.

**Limitations And Societal Impact:**

The authors have adequately addressed the limitations and potential negative societal impact of their work.

**Main Review:**

Originality: The question considered is new and very interesting. The proposed method is a novel combination of the three approaches: the surrogate index approach, the double machine learning approach and the dynamic treatment effect estimation approach. My concern is that, in terms of methodology innovations, there is not too much new outside of the three existing approaches.

Quality: As far as I checked, all claims are well supported. The theories and experimental results are explained very clearly.

Clarity: The paper is well written and I enjoyed reading it. Section 2.1 is particularly helpful in demonstrating the main idea. For the numerical experiments, I think it would be helpful to explain the five methods in a clearer way. Currently it is a bit confusing what the purple bar does.

Significance: The problem considered by the authors are important. I agree that treatment policies are often dynamic in practice. The solution to the problem is nice and strong; it has good theoretical properties and performs good empirically. Again, as mentioned above, my concern is that I don’t see too much new outside of the three existing methods.

**Time Spent Reviewing:**

3

---

> ### Author Response · Authors · 2021-08-10
> **Review response**
>
> We thank the reviewer for the elaborate review and comments and address each one in detail:
>
> 1) We view the identification strategy, which in the end decomposes that quantity of interest into components that can be estimated by variants of existing methods, a main non-trivial novelty of the paper, i.e. defining the target quantity of interest, in the presence of a dynamic treatment regime in the long-term data and deriving an estimand that identifies that quantity from the observed long-term and short-term datasets.
>
> 2) Moreover, even on the estimation side, we note that we significantly extend the surrogate estimation/identification approach of Athey et al, which was solely for the case of a single binary treatment. We offer here a Neyman orthogonal estimation and identification strategy which applies to any multi-dimensional continuous or binary treatments, which was important for our main application. This required a novel debiasing term for the surrogate setting as presented in Thoerem 3.2.
>
> 3) Finally, the novelty in Theorem 3.3 is that you can view the combination of the new variant from 3.2 and the estimator from Lewis and Syrgkanis as a single Neyman orthogonal moment estimator.

---

### Official Review · Reviewer_ZEEt · 2021-07-18

**Rating:** 7
**Confidence:** 3

**Summary:**

This paper develops a method for estimating cumulative effects of treatments when available data is limited to historical data of old treatments and short-term data of new treatments. A key challenge in this setting is that treatment assignments over time are correlated with each other, making it difficult to isolate the effects of individual treatments on the cumulative, long-term outcomes. This paper expands on previous methodology for this problem which make use of "surrogate indices" that mediate the effects of treatment on outcomes at each time step. Specifically, the authors estimate the isolated long-term reward of a treatment in the absence of future treatments. The methodology allows for continuous treatments and has double robustness properties (with respect to the surrogate index and surrogate score models). Semi-synthetic experiments show that the proposed dynamic adjustment approach lowers estimation error.

**Limitations And Societal Impact:**

The authors should at least make an explicit statement somewhere in the paper regarding limitations or that they do not foresee negative impact.

**Main Review:**

The topic of this paper is is very interesting and seems quite important: estimating long term effects of treatments from primarily historical data is important evidence for making early-stage decisions about new treatment policies. The authors state that their method combines (and extends) the surrogate index approach of Athey et al. (2020), double machine learning (Chernozhukov et al., 2018a), and dynamic effect adjustment (Lewis & Syrgkanis, 2020). I must admit that, of these techniques, I am only familiar with double machine learning. However, to the best of my understanding of the paper, the authors elegantly combine and extend these approaches to create a holistic solution to the long-term effect estimation problem.

As someone unfamiliar with the surrogate index approach of Athey et al. (2020), at times I had trouble following the methodology. For example, I do not believe the authors provide examples of surrogate variables, except to say that a single time period's outcome or "other measures... of a customer's trajectory" could be used as proxies. I think the investment firm example from the introduction could be fleshed out a little more to make clear what the proxies would be in that scenario.

Some of the organization in Sections 2 and 3 was confusing to me. In Section 2 the authors begin by describing the estimation strategy of Athey et al. (2020), then informally describe the assumptions required for identification before describing violations of these assumptions that would induce bias in the described estimation strategy. I think this section could be streamlined to more concisely state the key information that will be needed for problem setup and to understand the method. The target estimand and the assumptions required should be formally stated up front in Section 2. Further, the causal path assumption (that at each time step surrogates entirely mediate the effects of treatments on outcomes) is never formally stated. The graph in Fig 1 is very helpful for visualizing this, but it should be accompanied by the formal statements. Given this setup, the authors could then introduce the unadjusted surrogate index and clearly define the bias (right now the bias is shown in equation 3 but the authors do not draw much attention to this). The beginning of Section 3 and Section 3.3 were written in a way where it was much easier to follow. I think the authors should use this style of previewing what will be shown in each subsection in other parts of the paper as well.

The results in this paper are interesting and important. I think the paper could be greatly improved by cleanly and concisely introducing  the formal problem setup and previewing section content to help guide the reader.


UPDATE:
Thanks to the reviewers for their response. I remain positive about this work and will be maintaining my score. I definitely think the suggested improvements to clarity will help this work be appreciated by a broader audience (that includes myself!). I also think it should be possible for the authors to flesh out the hypothetical example(s) without revealing proprietary information---surely the proposed method is useful beyond its particular application at your organization! I genuinely believe these changes would take this paper from "good" to "great."

**Time Spent Reviewing:**

2.5

---

> ### Author Response · Authors · 2021-08-10
> **Review response**
>
> We thank the reviewer for the elaborate review and comments and address each one in detail:
>
> 1) We will expand more on the variables that we used as proxies in the empirical application. We need to be a bit vague due to proprietary reasons. We tried to have a more detailed description of the semi-synthetic dataset that mirrors the true dataset in Appendix C, but as you rightfully complain, it still doesn’t contain explicit proxy/surrogate variables. Some examples of these variables are: revenue streams from different parts of the corporate business, e.g. if the customer is buying multiple products from the corporation, then the revenue time series in the past few months for each of these products. Other measures of engagement with the company: for instance, revenue commitments signed through corporate contracts in the next six months. Usage of each of the products by the company (which can be different than revenue spend and can be indicative of growth). Time-varying demographic variables for the customer over time, such as customer size in terms of number of employees and several other non-time varying features, such as the industry that the customer belongs in, the geolocation of the customer.
>
> 2) We will revert the order and present the main setting and quantity of interest (i.e. the intro to Section 3) before section 2 and then go into the stylized example and how the existing methodology would fail and the source of bias. This seems a great idea. Also you are right, the causal graph in Figure 1 implies a conditional mean independence condition, that we heavily use in the identification argument (see proof of Theorem 3.1). We will explicitly state this conditional mean independence property that is implied by the causal graph (i.e. e.g. in the simple setting of no future treatments, simply that Y is mean-independent of T, conditional on the surrogates S_1). You are right that we do not explicitly state this mathematical condition that enables the identification strategy and is implicitly given in the causal graph. We mention it explicitly in words on line 90, where we say “Second, the only causal path from the treatment to the outcome goes through the surrogates.”, but we should provide a formal mathematical analogue of this, which is the conditional mean independence statement provided above (and it’s dynamic analogue).

---

### Official Review · Reviewer_SfcW · 2021-07-19

**Rating:** 8
**Confidence:** 3

**Summary:**

This paper proposes an estimation strategy for treatment effects on long term outcomes in a difficult setting with sequentially correlated treatments. They consider a certain case where unobserved values of the treatment show up in a small "new" dataset without long term outcomes recorded but where there is access to an "old" dataset with old treatments that include long term outcomes. Assuming certain nuisance functions (conditional expectations of various parts of the graphical model) can be estimated consistently with [insert favorite ML method here], they use Double ML to derive the asymptotic normality of their treatment effect estimators, as well as double robustness with respect to two of the nuisance functions.

**Ethical Concerns:**

None.

**Limitations And Societal Impact:**

Yes.

**Main Review:**

Nice work! In general the mathematical ingredients and how they piece together are clear but the writing is pretty unclear. Warning: I am fairly new to DML (a few months of reading) and have only seen it in simpler non-sequential settings eg basic ATE. However, all of my questions are not about the final application of DML / orthogonality itself but the setup leading up to it.

Some comments/questions

- how can A be px1 when the treatment vector is K dim ?

- This caused a lot of confusion for me when reading this paper, even though I could appreciate a lot of the piece-by-piece without knowing the answer to this question: could you describe the precise usage of the word "novel" in general in this work? The pharmaceutical example is estimation of causal effect of a new drug, but wouldn't this correspond to adding an extra treatment variable T,k+1 to the conditioning set of models like Y|T,S for a given timestep? That is Y|T1,...Tk,S vs Y|Y1,...,Tk+1,S?  And the models on the observational set are not estimated on such an expanded conditioning set? Or is it that you are estimating outcomes Ybar under some new treatment *value* T* in R^k such that T* not observed in observational data? Can you try to clarify here what precisely is meant by novel with an example? What assumptions are necessary on the nature of the new treatment?

- line 69: small off by one error in the M term sum, kappa should start at 1 instead of 0 as done later in line 165

- lines 94-96 since I had trouble understanding this description in words how correlated treatments cause bias but I understood the mathematical example starting at line 109. Maybe you can mention at 94-96 that this is demonstrated below?

- lines 135 (under eq 5): it would be useful to point out that gamma(I+B)S,t shows up earlier as the first term in eq2: g0(S,t). Likewise that eq 6 (line 137) recovers two of the terms in line 121 just above eq 1. Help the reader realize exactly where they saw these terms before.

- (important) After eq 6, a nice summary sentence, enumerated list, or formal Algorithm environment would be good to make sure people understand the algorithm. If I understand correctly it is  (Step one) First predict Y,t+1  | T,t+1, St as a linear function of T,T+1 and S,t that takes the specific form in (4) (Step 2) define adjusted outcome  Ybar,adj,t = Yt + Y,t+1 - alpha,t+1,Tt+1 which depends on the first step estimate through alpha (Step 3) Then predict this adjusted outcome from Tt,S,t-1 (Step 4) Finally the coef. in front of T,t (in a correctly specified model) is the causal effect theta0. Is it right?

- lines 142-144, what do you mean by "both" approaches, do you mean (one) surrogate index without dynamic adjustment, and (two) dynamic adjustment?

- line 164 to eq 7: you used E_o and E_e before defining them in 171

- sections 3.1 vs 3.2, DML / asymptotic normality: You mention in both sections that because of the orthogonal representation that normality results hold. For 3.1 (unadjusted) you say it in words, and for 3.2 (adjusted) you prove it (theorem 3.3). Does it trivially hold for section 3.1 (unadjusted) because of the proof in 3.2? Or just because it's the standard case already covered by DML?

- what about a small fully-synthetic experiment with a correlated earlier and later treatment where you are estimating the effect of the earlier treatment without adjustment and where you plot estimation bias vs strength of correlation (through either of the two routes: kappa or lambda*S)

- is there no public dataset that fits this problem setup that you can also include in addition to the semi-synthetic experiment?

- This paper is definitely missing a conclusion.

**Time Spent Reviewing:**

10

---

> ### Author Response · Authors · 2021-08-10
> **Review response**
>
> We thank the reviewer for the elaborate review and comments and address each one in detail:
> 1) In the paragraph right before the equations, on line 115 we mention: “focusing on a single scalar investment”. In this section, for expository purposes we only present the problem with the existing method in the case of a scalar treatment and not a vector treatment. Hence, why A is p x 1. In the general setup (e.g. in section 3.2), the treatments are allows to be k-dimensional.
>
> 2) We will expand on the discussion about novel treatments. The treatments in the new setting are allowed to be arbitrarily different as compared to the observational long-term setting. So you could have k new drugs being deployed in the last 6 months and k completely different old drugs that were deployed in your historical data. The only thing that needs to remain invariant is the relationship between the controls and the outcome and the next period controls with past period controls, in other words, how the outcome evolves when all treatments are set to zero (in absence of treatment), needs to be the same in the two settings. In the last paragraph of Section 2.1 (lines 142-151) we explain which parameters exactly need to remain invariant in the two settings, in the two linear Markovian case. More generally, the only invariance that needs to hold is the dynIR condition on line 138, which exactly expresses the notion that: the expected outcome in the absence of treatments evovles in the same manner in the two settings.
>
> 3) You are right!
>
> 4) Indeed, will be helpful to point that this is derived with mathematical rigor below.
>
> 5) Indeed, we will add pointers that clarify the relations of these equations and that in the end we recover what we wanted earlier.
>
> 6) We will add a short algorithmic description inline there. For space constraints we opted to present an algorithmic description of the estimation process in full detail in Appendix B. We will add a short summary of this inline and a pointer to Appendix B.
>
> 7) Will move definitions of E_e and E_o up.
>
> 8) Theorem 3.3 covers the case of what would be the normality statement analogue of theorem 3.2 and we omitted it due to space constraints, since it is a special case where the future treatments do not exist (e.g. are zero almost surely). Hence that Theorem 3.3 is a strictly more general normality result. We added one sentence to explain this at the end of section 3.1, but will expand.
>
> 9) That is a great experiment and we will add. Due to space constraints we could only add one of “synthetic” or “semi-synthetic” analysis. We have fully synthetic experiments in the appendix E, but we could not make them fit in the main text. Still those results do not contain the type of experiment you describe of the relationship between correlation strength vs bias and in particular the two types of autocorrelation and bias source! It is a great experiment to circle back to the simple two period example section.
>
> 10) Since the main application that we applied this to, was a corporate application, these datasets are typically private. Similarly, drug trial results with long-term and short-term data, would be hard to construct and not readily available and seems like a feat that would be a separate empirical application paper. In this work, we chose to focus on the methodological part and on the specific return-on-investment application that we applied this methodology to and for which we have experience of the practical complications that might arise from deploying the method in practice.

---

> > ### Comment · Reviewer_SfcW · 2021-08-25
> > **Thanks**
> >
> > Thanks for all of the clarification.
> >
> > Yes, I would be glad if you incorporate the correlation vs bias experiment. Even if you do not place it in main text, you can refer to it in the main text and place in appendix.
> >
> > Also, I think it is important to eventually follow through with your response to reviewer ZEEt. Please provide crystal-clear examples of surrogates in the main text when introducing them. I respect that there may be constraints proprietary-wise but this cannot get in the way of clarity in an open access academic conference paper.
> >
> > I learned a lot from this paper and would be glad to see this work accepted.

---

### Decision · Program_Chairs · 2021-09-27

**Decision:**

Accept (Poster)

**Comment:**

Thanks to the authors for this engaging and submission.  The reviewers were all quite positive about this work, and I'm happy to recommend acceptance.

One shared concern among the reviewers is clarity of the submission --- as reviewer ZEEt notes, sections 2 and 3 are a bit confusing in their presentation, and should be streamlined in the final version.  And as reviewer SfcW discussed, the incorporation of a new synthetic experiment will help understand the relationship between estimation bias and strength of correlation, even if placed in the appendix and referred to in the main text.